# The Role Played by Ferroptosis in Osteoarthritis: Evidence Based on Iron Dyshomeostasis and Lipid Peroxidation

**DOI:** 10.3390/antiox11091668

**Published:** 2022-08-27

**Authors:** Shaoyun Zhang, Jiawen Xu, Haibo Si, Yuangang Wu, Shengliang Zhou, Bin Shen

**Affiliations:** 1Orthopedic Research Institute, Department of Orthopedics, West China Hospital, Sichuan University, Chengdu 610041, China; 2Department of Orthopedics, The Third Hospital of Mianyang, Sichuan Mental Health Center, Mianyang 621000, China

**Keywords:** ferroptosis, osteoarthritis, iron dyshomeostasis, lipid peroxidation

## Abstract

Ferroptosis, a recently discovered regulated cell death modality, is characterised by iron-dependent accumulation of lipid hydroperoxides, which can reach lethal levels but can be specifically reversed by ferroptosis inhibitors. Osteoarthritis (OA), the most common degenerative joint disease, is characterised by a complex pathogenesis involving mechanical overload, increased inflammatory mediator levels, metabolic alterations, and cell senescence and death. Since iron accumulation and oxidative stress are the universal pathological features of OA, the role played by ferroptosis in OA has been extensively explored. Increasing evidence has shown that iron dyshomeostasis and lipid peroxidation are closely associated with OA pathogenesis. Therefore, in this review, we summarize recent evidence by focusing on ferroptotic mechanisms and the role played by ferroptosis in OA pathogenesis from the perspectives of clinical findings, animal models, and cell research. By summarizing recent research advances that characterize the relationship between ferroptosis and OA, we highlight avenues for further research and potential therapeutic targets.

## 1. Introduction

Osteoarthritis (OA) is the most common degenerative joint disease that affects 7% of the global population and more than 500 million people worldwide [1]. Due to population ageing, prolonged life expectancy, increasing obesity, and other causes, the incidence and prevalence of OA are gradually increasing [1,2]. OA is a whole-joint disease involving the cartilage, synovium, subchondral bone, infrapatellar fat pad, ligaments, meniscus, capsule, and periarticular muscles [2]. Its aetiology and pathogenesis are complex and have not yet been clarified, but the key role played by cartilage degeneration in OA has been recognized [3]. Chondrocytes are the only cell type in cartilage and are critical for the biogenesis and maintenance of the extracellular matrix, which is composed of type II collagen (COL2), hyaluronic acid, and chondroitin sulfate proteoglycan [4]. Loss of homeostasis in cartilage contributes to OA development when the chondrocyte synthetic capacity is overwhelmed by processes that promote matrix degradation.

Oxidative stress plays an important role in OA, causing inflammation and matrix degradation in joints. Reactive oxygen species (ROS) production and subsequent lipid peroxidation are related to the antioxidant capacity of chondrocytes, playing key roles in cartilage degradation and chondrocyte death [5,6]. Lipid peroxidation, which often leads to lipid hydroperoxide formation, occurs in response to oxidative stress. In recent years, a newly discovered form of regulated cell death named ferroptosis, which is characterized by the iron-dependent accumulation of lipid hydroperoxides that reach lethal levels, has been reported to be associated with OA pathogenesis [7,8,9]. Yao et al. [8] first indicated that chondrocytes underwent ferroptosis under inflammatory and iron overload conditions and that ferroptosis contributed to the progression of OA in vivo and promoted matrix metalloproteinase (MMP)-13 expression while inhibiting COL2 expression in chondrocytes cultured in vitro. Miao et al. [9] found that iron accumulated in cartilage and synovial fluid during OA progression and that the expression of biomarkers of the peroxidation defence system, including glutathione peroxidase (GPX) 4 (GPX4) and glutathione (GSH) levels, was decreased in these patient samples. Moreover, as a characteristic change in ferroptosis, morphological changes in mitochondria have also been observed in OA cartilage by transmission electron microscopy, indicating that ferroptosis is closely associated with OA.

As these data suggest that OA may share similar pathological characteristics with ferroptosis in terms of iron dyshomeostasis and lipid peroxidation, a review of the role played by ferroptosis in OA development is important, and therefore, we summarize the latest evidence, focusing on ferroptotic mechanisms and the role that ferroptosis plays in OA pathogenesis from the perspectives of clinical findings, animal models, and cell research. By summarizing recent advances in research that characterize the relationship between ferroptosis and OA, we highlight avenues for further research and potential therapeutic targets for this disease.

## 2. Main Characteristics of Ferroptosis

Ferroptosis is distinct from apoptosis, autophagy, and necrosis in terms of cellular morphology, biochemistry, and genetics [7]. The morphological features of ferroptotic cells manifest as an aberrant mitochondrial ultrastructure, including a reduction in mitochondrial volume, an increase in mitochondrial membrane density, and the disappearance of mitochondrial cristae in ferroptotic cells, as indicated by electron microscopy [7,10]. Iron accumulation and lipid peroxidation are increasingly recognized as central mediators of ferroptosis. The subsequent formation of lipid hydroperoxides and a diminished antioxidant system directly leads to ferroptosis [11]. Furthermore, a genetic network that differs from that of other cell death modalities governs ferroptosis [7].

### 2.1. Iron Homeostasis and Ferroptosis

Systemic iron homeostasis is maintained by balancing iron supply, utilisation, and losses [12]. Iron is mainly consumed for erythrocyte generation, and it enters the circulatory system through reticuloendothelial macrophages that salvage iron from aged erythrocytes at a rate of 20–25 mg per day and from duodenal enterocytes, which absorb 1–2 mg dietary iron per day [12]. The absorbed iron is transported into enterocytes by divalent metal-ion transporter 1 (DMT1) [13] and is then exported into the bloodstream by ferroportin (FPN), which functions with the ferroxidase hephaestin, which oxidizes ferrous iron (Fe^2+^) to ferric iron (Fe^3+^), the form that binds transferrin (Tf) [14,15]. The Tf-bound iron circulates throughout the body to deliver iron to peripheral tissues [14]. Iron is lost at a rate of 1 mg per day mainly through sloughing of epithelial cells and bodily fluid loss [12]. Hepcidin, a key regulator of systemic iron homeostasis, is a small circulating peptide produced mainly by hepatocytes and can bind FPN on enterocytes, macrophages, and other cells to trigger FPN degradation and block iron efflux [16].

In the circulatory system, Tf-bound Fe^3+^ is taken up by cells through receptor-mediated endocytosis after Tf binds to the membrane protein transferrin receptor (TfR) 1 (TfR1) [17]. In the low pH environment of endosomes, Fe^3+^ is released from Tf-TfR1 complexes and reduced to Fe^2+^ through the ferrireductase activity of six-transmembrane epithelial antigens of prostate 3 [18]. Then, Fe^2+^ in the endosome is imported into the cytoplasm via DMT1 [18]. Most intracellular iron is bound to ferritin, an iron storage protein complex consisting of ferritin light chain (FTL) and ferritin heavy chain 1 (FTH1) [19]. A small amount of unbound iron comprises the labile iron pool, which plays a role in regulating iron homeostasis [20]. FPN-mediated iron ion efflux functions in combination with the multicopper ferroxidase hephaestin, which oxidizes Fe^2+^ to Fe^3+^, which binds Tf [14,15].

Excessive iron can lead to ROS production through the Fenton reaction and activation of iron-containing enzymes (such as lipoxygenase) that promote lipid peroxidation and lead to ferroptosis. Thus, ferroptosis is promoted by increasing iron absorption, reducing iron storage, or limiting iron efflux, and therefore, iron chelators can prevent ferroptosis [21]. Feng et al. [22] used an anti-3F3 ferroptotic membrane antibody (3F3-FMA) to detect ferroptotic cells, discovering that 3F3-FMA is a TfR1 antigen; hence, they concluded that TfR1 accumulation on the cell surface is a feature of ferroptosis. In researching baicalin-triggered ferroptosis in vitro and in vivo, Kong et al. [23] found higher intracellular chelated iron levels after FTH1 overexpression in bladder cancer cells, indicating that baicalin-induced ferroptosis was accelerated by downregulating FTH1 expression. Bao et al. [24] found ferroptosis phenotypes in the brains of Alzheimer’s disease (AD) model mice, and in these mice, ferroptosis was induced by downregulating FPN expression. In contrast, FPN overexpression in the hippocampus partially attenuated the ferroptosis rate and ameliorated memory impairment in the AD model mice. Indeed, directly increasing the exogenous iron supply, such as through ferric ammonium citrate (FAC), enhanced erastin-induced ferroptosis, which was inhibited by iron chelators such as deferoxamine (DFO), thereby reducing iron overload [7].

### 2.2. Lipid Peroxidation and Ferroptosis

Lipid peroxidation, which was thoroughly reviewed by Ayala et al. [25], is a process in which oxidants such as free radicals or ROS attack lipids containing carbon–carbon double bond(s), especially polyunsaturated fatty acids. Overall, lipid peroxidation consists of three steps: initiation, propagation, and termination. Once lipid peroxidation is initiated, chain reactions continue until termination products are produced [26]. The main primary products of lipid peroxidation are lipid hydroperoxides, and the main secondary products are malondialdehyde (MDA) and 4-hydroxynonenal (4-HNE) [25]. Due to its high reactivity and reliability, MDA is an oxidative stress biomarker commonly used in clinical situations [27]. 4-HNE is currently considered a major bioactive marker of lipid peroxidation and a signalling molecule involved in the regulation of transcription factors sensitive to stress, such as nuclear factor erythroid 2-related factor 2 (Nrf2), in cell proliferation, differentiation, and death [25]. In ferroptosis, MDA and 4-HNE are reliable markers of oxidative stress-induced lipid peroxidation in cancer [28], AD [29], and acute lung injury [30].

In general, ferroptosis is triggered when lipid peroxidation production overwhelms the antioxidant-buffering capacity of cellular antioxidant systems. At least three antioxidant systems control ferroptosis: the cyst(e)ine/GSH/GPX4 axis, ferroptosis suppressor protein 1 (FSP1)/coenzyme Q10 (CoQ10) axis, and the cyclohydrolase 1/tetrahydrobiopterin /dihydrofolate reductase axis [31]. The cyst(e)ine/GSH/GPX4 axis is the most frequently targeted pathway to trigger the ferroptosis cascade [31]. GSH is an important intracellular antioxidant; cystine is the raw material for GSH synthesis, and the cystine/glutamate antiporter system x_c_^−^ on the cell membrane typically mediates the exchange of extracellular cystine and intracellular glutamate [32]. GPX4 is a GSH-dependent enzyme that converts reduced GSH to oxidized glutathione (GSSG) and simultaneously reduces lipid hydroperoxides to the corresponding lipid alcohols or free hydrogen peroxide (H_2_O_2_) to water [31]. Disruption of system x_c_^−^-mediated cystine uptake or GSH depletion leads to the inactivation of GPX4, allowing lipid peroxides accumulation, which triggers ferroptosis. FSP1 localizes to the plasma membrane and functions as a NAD(P)H-dependent oxidoreductase capable of reducing CoQ10, which can trap lipid peroxyl radicals, thereby suppressing lipid peroxidation and ferroptosis [33].

Downregulation of antioxidant system activation has been reported to be associated with ferroptosis. Studying a genetically engineered mouse model, Badgley et al. [34] reported that deletion of solute carrier family 7, member 11 (SLC7A11, a system x_c_^−^ subunit) induced tumour cell ferroptosis and doubled median survival compared to a vehicle treatment, and mice treated with the antioxidant N-acetyl cysteine (NAC) exhibited restoration of baseline survival and elimination of tumour responses, supporting a link to cyst(e)ine metabolism. Yang et al. [35] found that inhibition of GPX4 by DL-buthionine-S,R-sulfoximine (BSO, a GSH-depleting reagent) sensitized cells to death induced by 12 divergent compounds, whereas activation of GPX4 by cDNA overexpression rescued cells from the lethality induced by these compounds, indicating that ferroptosis is mediated through a GPX4-regulated pathway. Studying hundreds of cancer cell lines, Bersuker et al. [36] found that FSP1 expression was positively correlated with ferroptosis defence and that FSP1 inhibited ferroptosis by reducing CoQ10 levels in cultured lung cancer cells and mice carrying tumour xenografts.

## 3. Potential Association between Ferroptosis and OA: Clinical Findings

Clinical findings supporting the potential association between ferroptosis and OA are summarized in Table A1, with a schematic representation in Figure 1.

### 3.1. Iron Dyshomeostasis

Iron dyshomeostasis in clinical haemophilic arthropathy and inherited haemochromatosis arthropathy was thoroughly reviewed by Sun et al. [37]. Iron accumulation and related iron dyshomeostasis have been found in patients with primary OA. Yazar et al. [38] found that the iron ion level in the synovial fluid of OA sites in OA patients was significantly increased compared with that in healthy subjects. Miao et al. [9] found that the iron level in synovial fluid was positively correlated with OA severity. Moreover, Fe^2+^, Fe^3+^, and total iron concentrations were all significantly higher in the cartilage in OA-damaged areas than in undamaged areas, indicating that iron had accumulated in the cartilage during OA progression. Moreover, iron deposition was also found in the synovia of patients with OA [39].

In the blood circulatory system, serum iron and ferritin are indicators of total body iron store level. Two-sample Mendelian randomisation analyses showed that serum iron was positively associated with an increased risk of unspecified OA in males [40], and a similar correlation was found in females with OA [41]. Performing a genome-wide association study and pathway analyses, Liu et al. [42] reported that iron ion transport pathways were significantly associated with knee OA in African Americans. Nugzar et al. [43] evaluated the association of serum ferritin level with cartilage damage severity, as assessed by arthroscopy in patients with knee OA, and found that the serum ferritin level increased with cartilage damage severity, and these results were independent of age, sex, body mass index, and C-reactive protein level, suggesting that ferritin may be actively involved in the progression of cartilage damage in patients with symptomatic knee OA. Kennish et al. [44] found that higher levels of serum ferritin were positively correlated with worsening Kellgren–Lawrence stage in the total OA population, particularly in men with OA.

In addition, iron intake seems to be associated with the progression of OA. Wu et al. [45] found a U-shaped association between iron intake and the knee OA progression. They concluded that appropriate iron intake was advisable for preventing OA progression, whereas excessive or deficient iron intake increased the risk of OA progression.

### 3.2. Lipid Peroxidation

The level of oxidative stress is represented by MDA and 4-HNE and is closely associated with OA progression. In 2003, Grigolo et al. [46] evaluated the degree of lipid peroxidation in synoviocytes of patients with OA and controls by colorimetric assay and found increased levels of MDA and 4-HNE in the synoviocytes of the OA group compared with those in the control group. They hypothesized that this peroxidation process might have been due to the action of nitric oxide (NO) secreted by chondrocytes, which led to higher radical levels in OA. Increased levels of 4-HNE were also found in the synovial fluids of patients with OA [47]. Moreover, Shah et al. [48] performed immunohistochemical staining and detected MDA and 4-HNE in OA tissues and weak immunostaining of the cartilage surface in sections of normal cartilage. Performing a thiobarbituric acid reactive substance assay, Gavriilidis et al. [49] also found higher levels of MDA in OA cartilage than in control cartilage.

Downregulation of antioxidant system activity has been detected in patients with OA. Regan et al. [50] detected reduced GSH and GSSG levels in the synovial fluid of 27 OA patients compared with those in 12 patients undergoing knee arthroscopy with macroscopically intact cartilage. Maneesh et al. [51] found reduced GSH and GPX levels in the plasma of OA patients compared with those in healthy controls. Miao et al. [9] found decreased GPX, GSH, and GSH/GSSG levels in OA cartilage. Moreover, they performed RNA sequencing to evaluate transcriptome data obtained for OA cartilage and undamaged cartilage and found that the expression levels of GPX4 and solute carrier family 3, member 2 (SLC3A2, a system x_c_^−^ subunit) were lower in the OA cartilage. These results were consistent with those of a ferroptosis assay [7].

Vitamin E is a well-known lipophilic antioxidant that reduces cell lipid peroxide levels and prevents ferroptosis [52]. Notably, the vitamin E level is negatively related to OA progression. Specifically, Sutipornpalangkul et al. [53] found that the vitamin E concentration in synovial fluid was inversely related to primary knee OA severity in 32 patients, indicating that oxidative stress increased as the clinical severity of OA increased. A similar study later confirmed this result [54]. Regarding the therapeutic effect of vitamin E supplementation, Bhattacharya et al. [55] conducted a cohort study in which the levels of antioxidant enzymes, such as GPX and MDA in plasma, were estimated in 40 healthy individuals (control group) and in 40 OA patients 50–70 years old before and after 3 months of vitamin E supplementation; the patients were divided into group I (no supplementation) and group II (200 mg/day vitamin E supplementation. Decreased GPX and increased MDA levels were found in the OA patients without vitamin E supplementation compared with those in the control individuals, and these levels were significantly decreased in the OA patients after vitamin E supplementation.

## 4. Potential Association between Ferroptosis and OA: Animal Models

Animal models supporting the potential association between ferroptosis and OA are summarized in Table A2, with a schematic representation in Figure 2.

### 4.1. Iron Dyshomeostasis

In recent years, a positive correlation between iron overload and OA has been reported. Burton et al. [56] found higher levels of iron in articular cartilage and infrapatellar fat pads of an iron-overloaded group compared to those in the control group. Excess iron worsened knee OA, as determined by both micro-computed tomography and a histologic scoring system. Moreover, exogenous iron altered the expression of iron trafficking proteins and certain cytokines, and affected structural cartilage components. With iron-overloaded and/or destabilization of the medial meniscus (DMM)-established OA mouse models, Jing et al. [57] found higher levels of iron in cartilage and synovial tissue of the iron-overloaded DMM-induced group than in either the iron-overloaded group or the DMM-induced group. Increased expression of a disintegrin and metalloproteinase with thrombospondin motifs 5 (ADAMTS-5) and MMP-13 and higher Osteoarthritis Research Society International scores were observed in the iron-overloaded DMM-induced group than in the DMM-induced group without iron overload. These results suggest a close relationship between iron overload and OA.

Iron dyshomeostasis has also been reported in experimental OA models. Radakovich et al. [58] found that obese guinea pigs exhibited an increase in the expression TfR1 in cartilage that was more than two-fold higher than that in calorie-restricted guinea pigs, and this increase was associated with the development of spontaneous knee OA. Luo et al. [59] explored changes in the synovial fluid proteome in rabbit models of anterior cruciate ligament transection (ACLT)-induced OA and found that the Tf level was increased and that the FTH1 level was decreased in the model group compared with the normal group.

Reports on experimental OA models have indicated that iron chelators can prevent OA progression. To determine whether reduced iron level induced by pharmacologic iron chelation with DFO affected the development and/or severity of cartilage lesions in a primary OA model, Burton et al. subcutaneously injected DFO into Dunkin–Hartley guinea pigs [60]. They found that the number of OA-associated cartilage lesions was reduced in the knees of the DFO-treated animals, with chondrocyte hypocellularity identified as a key histologic difference between groups, suggesting that iron chelation delayed primary OA progression in this animal model.

### 4.2. Lipid Peroxidation

Lipid peroxidation has been identified in experimental OA models, similar to patients with OA. Karakurum et al. [61] reported that serum MDA levels were positively correlated with degeneration severity in rabbit models of ACLT-induced OA. Yang et al. [62] found increased MDA and GSSG levels and decreased GSH and GPX levels in the serum of ACLT-induced OA rat models compared to controls. Goranov et al. [63] and Chang et al. [64] found similar results in the same OA model of dogs and obese rats. Gladkova et al. [65] and Zubavlenko et al. [66] found increased MDA and lipid peroxide levels in the serum of ACLT-induced OA rat models. Bai et al. [67] found increased MDA and decreased GSH levels in the serum of rat models of OA established by DMM compared to control rats. Bai et al. [68] observed increased MDA and decreased GSH levels in the synovium and articular cartilage of ACLT-treated rabbits compared to control rabbits, and Danshen reversed OA progression. Aulin et al. [69] and Yang et al. [62] found an increased 4-HNE level in the cartilage of an OA group. Shi et al. [70] found higher levels of 4-HNE in the synovial fluid and cartilage of dogs with ACLT-induced OA than in the sham group. Moreover, this was the first group to report that intraarticular injection of 4-HNE into dog stifle joints induced cartilage lesions and expression of MMP-13, ADAMTS-5, and cyclooxygenase-2 (COX-2). Similarly, Zhou et al. reported increased MDA and decreased GPX4 levels in ACLT-induced OA cartilage [71]. Qiu et al. [72] found decreased GSH and GPX levels in the cartilage of an OA model established by medial meniscus resection. These results indicate that lipid peroxidation is associated with OA pathophysiology in vivo.

In addition to models established by surgically induced OA, monosodium iodoacetate (MIA)-induced OA models and other OA models have shown lipid peroxidation. Pathak et al. [73] and Abdel Jaleel et al. [74] found increased MDA and decreased GSH levels in the plasma of MIA-induced OA rats compared to control rats. Similarly, Huang et al. reported decreased GSH abundance in serum [75]. Fusco et al. [76] found an increased MDA level and decreased GSH and GPX levels in the serum of MIA-induced OA rats compared to control rats. Yabas et al. [77] and Ragab et al. reported the same results [78]. Ajeeshkumar et al. [79] also found decreased GSH abundance in the joint tissues of MIA-induced OA rats compared to control rats. Ma et al. [80] reported increased MDA and decreased GPX levels in the serum of rats with complete Freund’s adjuvant (CFA)-induced OA compared to control rats. Quercetin [72], cashew nuts [76], zinc [75], type III collagen [74], proteoglycans [79], and platelet-rich plasma [78] reversed OA-associated oxidative damage by inhibiting lipid peroxidation.

As previously mentioned, antioxidants can prevent OA development in experimental models. In 2002, Kruz et al. [81] investigated the influence of dietary vitamins and selenium on the progression of OA caused by varus deformity-induced mechanical overload of the medial tibial plateau and the expression of antioxidative enzymes in model mice. They found that a special diet decreased OA incidence and increased the expression of GPX in the cartilage, synovium, and serum. However, the component of the special diet that played a major role in reversing OA is unknown.

### 4.3. Iron Dyshomeostasis and Lipid Peroxidation

In recent years, iron dyshomeostasis and lipid peroxidation have been reported in OA animal models, and iron chelators or antioxidants have prevented OA development. In mouse models with ACLT-induced OA, Miao et al. [9] found that the expression of GPX4 and FTH1 was decreased, which was consistent with a ferroptosis assay. Intra-articular injection of DFO or ferrostatin-1 (Fer-1, an antioxidant) twice per week for eight consecutive weeks attenuated OA development in ACLT-treated mice by inhibiting chondrocyte ferroptosis. Moreover, Miao et al. [9] reported that intra-articular injection of a GPX4 short hairpin RNA fragment cloned in an adeno-associated virus (AAV) (AAV-shGpx4) two weeks before ACLT surgery accelerated OA progression. This result shows that the downregulation of antioxidant system activation is positively correlated with OA development. Yao et al. [8] also reported decreased GPX4 expression in cartilage OA induced by DMM, and intra-articular injection of Fer-1 reversed cartilage degeneration. Lv et al. [82] reported increased MDA and Fe^2+^ levels in synovial fluid and decreased GPX4 expression in OA cartilage in DMM-induced rat models. They also found that downregulation of staphylococcal nuclease domain containing 1 (SND1) expression upregulated GPX4 expression in the cartilage of DMM-treated rats, inhibiting ferroptosis and reducing OA progression.

Furthermore, Guo et al. [83] found that intra-articular injection of erastin (an inducer of ferroptosis) induced cartilage COL2 loss and significantly increased the number of MMP-13-positive cells, indicating that erastin induced OA-like changes in chondrocytes and promoted OA development. Furthermore, intra-articular injection of DFO-alleviated DMM- and erastin-induced OA development.

## 5. Potential Association between Ferroptosis and OA: Cell Research

Cell researches supporting the potential association between ferroptosis and OA are summarized in Table A3 and Table A4, with a schematic representation in Figure 3.

### 5.1. Iron Dyshomeostasis

Iron overload disrupts cellular iron homeostasis, which compromises the functional integrity of chondrocytes and leads to oxidative stress and ferroptosis. In 1982, Kirkpatrick et al. [84] found that exogenous Fe^3+^, Fe^2+^, or ferritin inhibited proteoglycan synthesis, indicating a possible pathway whereby cartilage is susceptible to destruction. Karim et al. [85] treated chondrocytes with exogenous FAC to mimic iron overload in vitro and found increased FTH1 expression and significantly decreased expression of hepcidin, FPN, TfR1, and TfR2. Furthermore, high doses of FAC increased labile iron and ROS levels, decreased COL2 production, disrupted the cell cycle, and increased the cell death rate compared with untreated controls. Jing et al. [86] found increased intracellular iron and ROS levels and higher expression of MMP-3, MMP-13, and ADAMTS-5 in FAC-treated chondrocytes. Moreover, they detected mitochondrial dysfunction. All these effects were reversed by cotreatment with the calcium chelator BAPTA acetoxymethyl ester. A dose-dependent decrease in the expression of COL2 and SRY (sex-determining region Y)-box 9 was also detected in FAC-treated chondrocytes [57]. Ohno et al. [87] examined the effects of excess iron on the differentiation and mineralization of cultured chondrocytes and ATDC5 cells. They found that FAC inhibited calcium deposition and increased iron accumulation. FAC inhibited the expression of MMP-13 and enhanced the expression of FTH1 and FTL. These results suggest that iron overload might cause osteopenia and arthritis by inhibiting chondrocyte differentiation and mineralisation.

Interleukin-1beta (IL-1β) enhanced iron influx and attenuated iron efflux in OA chondrocytes by upregulating TfR1 and DMT1 expression and downregulating FPN expression. In addition, downregulating DMT1 expression reversed the increase in MMP-3 and MMP-13 expression and the decrease in COL2 and inducible NO synthase (iNOS) expression induced by IL-1β [57]. DFO reversed the increased expression of MMP-3 and MMP-13 induced by IL-1β [57]. Tchetina et al. [88] investigated the effects of DFO on collagen cleavage, inflammation, and chondrocyte hypertrophy in relation to energy metabolism-related gene expression in OA articular cartilage. They found that collagen cleavage was frequently suppressed by DFO. Furthermore, DFO downregulated the expression of MMP-1, MMP-13, IL-1β, tumor necrosis factor-alpha (TNF-α), and type X collagen, a marker of chondrocyte hypertrophy. In contrast, the expression of genes associated with the mitochondrial Krebs cycle (also known as the tricarboxylic acid cycle), adenosine monophosphate-activated protein kinase, hypoxia-inducible factor (HIF)-1alpha, and COL2 was upregulated. Lactoferrin is a naturally occurring iron chelator, and Rasheed et al. [89] found that lactoferrin treatment inhibited COX-2 expression and prostaglandin E2 (PGE2) production induced by IL-1β in human OA chondrocytes, indicating Lf anti-arthritic activity.

### 5.2. Lipid Peroxidation

Lipid peroxidation leads to OA-like changes in cartilage and chondrocytes. Morquette et al. [47] incubated OA cartilage explants with 4-HNE and found that 4-HNE accelerated COL2 degradation by activating MMP-13. In isolated OA chondrocytes, 4-HNE decreased COL2 expression and increased MMP-13 expression. Vaillancourt et al. [90] then confirmed that 4-HNE induced COX-2 protein and mRNA expression with accompanying increases in PGE2 production, which was reversed by the iNOS inhibitor N-iminoethyl-L-lysine in human OA chondrocytes and cartilage explants [91]. Benabdoune et al. [92] found that human OA chondrocytes treated with either IL-1β or 4-HNE resulted in increased COX-2, PGE2, and MMP-13 and decreased GSH expression and that these effects were reversed by resolvin D1 treatment.

Elmazoglu et al. [93] observed increased ROS and 4-HNE levels and decreased GPX and COL2 levels in cultured human OA chondrocytes. Yao et al. [94] also found increased intracellular ROS and decreased GPX levels in human OA chondrocytes, which were restored to basal levels by nifedipine, which activated the Nrf2 pathway. In addition, several inducing agents lead to OA-like changes in chondrocytes by increasing the lipid peroxidation rate and diminishing the effect of the peroxidation defence system. Hosseinzadeh et al. observed intracellular accumulation of ROS and increased MDA in IL-1β-induced chondrocytes, as well as decreased expression of GPX1 and GPX4 [95,96]. Zuo et al. [97] and Yin et al. [98] found increased intracellular ROS and decreased GPX levels in the same model cells. Zhu et al. [99] reported increased intracellular ROS and MDA levels and decreased expression of GSH in IL-1β-induced OA-like chondrocytes and showed that circ_0136474 activity or upregulated miR-766-3p expression attenuated oxidative injury. Wang et al. [100] reported increased intracellular ROS and decreased expression of GSH in TNF-α-treated chondrocytes. Moreover, chondrocytes incubated with H_2_O_2_ exhibited decreased GSH concentrations, and NAC treatment followed by activation with H_2_O_2_ significantly increased GSH concentrations compared with the effect of H_2_O_2_ activation alone [101]. Guo et al. [102] found increased ROS levels and decreased GSH and GPX expression in H_2_O_2_-treated chondrocytes, while Zhang et al. [103] found an increase in the cellular lipid peroxidation rate and a decreased GSH/GSSG ratio. Hence, exposure to oxidative stress-promoting treatments enhanced stress resistance by increasing the GSH content and GSH/GSSG ratio in chondrocytes [104]. Qiao et al. found increased ROS and MDA levels and decreased Nrf2 and hemeoxygenase-1 expression in MIA-induced chondrocytes [105]. A high glucose (HG) concentration can disrupt chondrocyte homeostasis and contribute to OA pathogenesis. Hosseinzadeh et al. [106] reported increased intracellular ROS and MDA levels, as well as decreased GPX1, GPX3, and GPX4 expression in chondrocytes with HG-mediated oxidative stress. Advanced glycation end products (AGEs) play vital roles in catabolic metabolism in cartilage of OA [107]. Chondrocytes treated with AGEs demonstrated increased intracellular ROS levels, increased MMP-1, MMP-3, and MMP-13 expression, and decreased GSH expression [108]. Ginger extract [95], diallyl disulfide [96], icariin [97], etomidate [98], nintedanib [100], plumbagin [102], four-octyl itaconate [103], lutein [105], and atorvastatin [106] have been reported to reverse this oxidative damage.

In 2000, Tiku et al. [109] found that exposure of chondrocytes to H_2_O_2_ resulted in oxidative damage to the cell matrix. However, vitamin E administered at physiological concentrations significantly diminished the release of labelled matrix components from activated chondrocytes. Furthermore, vitamin E diminished aldehyde–protein adducts formation in extracts of activated cells, which suggested that vitamin E played an antioxidant role in preventing protein oxidation. Tiku et al. [110] further demonstrated that chondrocyte-derived MDA mediated cartilage collagen oxidation, and glucosamine prevented in vitro collagen degradation in chondrocytes by inhibiting advanced lipid oxidation- and protein oxidation-related reactions [111,112]. Nishimura et al. [113] confirmed that lipid peroxidation products such as oxidized low-density lipoprotein are involved in cartilage matrix degradation. Cheng et al. [114] reported that GSH-loaded hydrogels prevented ageing chondrocytes from undergoing oxidative damage by increasing catalase activity, downregulating inflammatory gene expression, and decreasing the cell death rate.

### 5.3. Iron Dyshomeostasis and Lipid Peroxidation

In 2006, Dombrecht et al. [115] were the first to find that the addition of Fe^2+^ enhanced lipid peroxidation in tertiary butyl hydroperoxide (TBHP)- or H_2_O_2_-treated chondrocytes. In recent years, numerous studies have reported an association between OA and iron dyshomeostasis and lipid peroxidation. Jing et al. [116] investigated the roles played by iron homeostasis and iron overload-mediated oxidative stress in OA chondrocytes. They found that IL-1β and TNF-α disrupted iron homeostasis in chondrocytes by upregulating TfR1 and downregulating FPN expression. IL-1β combined with FAC induced enhanced MMP-3, MMP-13, and ADAMTS-5 expression in chondrocytes, which was reversed by DFO or NAC treatment of the FAC-treated chondrocytes. Yao et al. [8] found increased intracellular ROS and lipid ROS levels and decreased GPX4 and SLC7A11 expression in IL-1β- or FAC-induced OA-like chondrocytes via regulation of the Nrf2 antioxidant system. Moreover, erastin, the best-characterised classical ferroptosis inducer, promoted MMP-13 expression while inhibiting COL2 expression in chondrocytes. Mo et al. [117] reported increased intracellular Fe^2+^ and MDA levels and TfR1 expression and decreased GSH, GPX4, and SLC7A11 expression in IL-1β-treated mouse chondrogenic cells (ATDC5). Moreover, identical results, increased intracellular Fe^2+^ and MDA levels and decreased GPX4 expression, were reported by Lv et al. [82], who also found that the RNA-binding protein SND1 promoted GPX4 degradation by destabilizing heat shock protein family A member 5 (HSPA5) mRNA and suppressing HSPA5 expression, thus promoting the ferroptosis of OA chondrocytes. Guo et al. [83] found increased intracellular Fe^2+^, MDA, ROS, and lipid ROS levels and decreased GPX4 and SLC7A11 expression in IL-1β- or erastin-induced OA-like chondrocytes, which was reversed by DFO treatment via activation of the Nrf2 antioxidant system. Miao et al. [9] also reported the same results in TBHP-treated chondrocytes, and they suggested that GPX4 knockdown led to OA-like changes through the phosphoinositide 3-kinases-Akt and mitogen-activated protein kinase (MAPK) pathways. Further research by Zhou et al. [71] confirmed that HIF-2alpha was a central mediator in the D-mannose-induced ferroptosis resistance of chondrocytes. These cells were rescued from death by Fer-1 [8,9,71,82,83,117], indicating that these cells had undergone ferroptosis.

## 6. Discussion

Ferroptosis is an iron-dependent cell death modality characterised by lipid peroxidation [7]. In recent years, numerous studies have demonstrated the important role played by ferroptosis in OA. However, a summary of the evidence used to characterise ferroptosis in OA due to iron dyshomeostasis and lipid peroxidation has been lacking. In this review, we summarise recent evidence from the perspective of clinical findings, animal models, and cell research. Clinical observations revealed that iron accumulation and altered expression of iron-related proteins are common in the serum or plasma, synovial fluid, synovium, and cartilage of OA patients. Moreover, the increased degree of lipid peroxidation represented by MDA and 4-HNE levels and the downregulation of antioxidant system activation represented by GSH and GPX expression levels are closely associated with OA progression. Furthermore, we observed iron accumulation and lipid peroxidation in OA models and osteoarthritic changes in iron-overload models. These changes were reversed by iron chelators or antioxidants in vivo or in vitro. All of the evidence indicates that ferroptosis is closely related to OA.

Other ferroptosis pathways, such as the FSP1/CoQ10 axis, have been shown to play roles in OA. CoQ10 is an antioxidant that participates in energy production in the human body. Chang et al. [118] conducted a case-control study to investigate the association of CoQ10 in plasma with OA. They found that elderly patients with OA presented with a slightly significantly lower CoQ10 level than subjects without OA. Using an experimental model of rat OA induced by intra-articular injection of MIA into a knee, Lee et al. [119] found that CoQ10 exerted a therapeutic effect on OA, as indicated by pain suppression and cartilage degeneration, by inhibiting the expression of inflammation mediators. Li et al. [120] investigated whether CoQ10 suppresses catabolic responses of IL-1β-induced chondrocytes and found that CoQ10 suppressed MMP-3, MMP-9, and MMP-13 production and markedly inhibited IL-1β-induced MAPK pathway activation in rat chondrocytes. These results provide insight into the potential mechanisms by which CoQ10 protects against cartilage degeneration in patients with OA.

However, some studies have reported different findings regarding lipid peroxidation and antioxidant systems in OA. For example, Ostalowska et al. [121] found that, compared to control subjects, patients in both primary and secondary knee OA subgroups presented with significantly increased activity of all antioxidant enzymes and glutathione transformation enzymes in synovial fluid. Mathy-Hartert et al. [122] reported that GPX activity and gene expression were both increased in a dose- and time-dependent manner in IL-1β-treated bovine chondrocytes. Studying ACLT- and medial meniscectomy-induced OA in rats, Tsai, et al. [123] observed that intra-articular injection of sulfasalazine (a system x_c_^−^ inhibitor) reduced the glutamate content in synovial fluid and the GSH level in chondrocytes and significantly attenuated knee swelling and cartilage destruction in knee OA. These results may have been related to the redox imbalance and enhanced antioxidant system activity in the early OA stage, while in the late OA stage, the antioxidant system may have been overwhelmed with excessive ROS, which it could not eliminate, leading to the antioxidant system breakdown [124].

Evidence has shown temporal and spatial changes in iron homeostasis and lipid peroxidation. Brodziak-Dopierała et al. [125] found significantly different levels of iron in various components of the knee joint in patients with OA. The highest iron content was found in the femoral bone portion of the knee joint followed by the meniscus, and the lowest iron content was found in the tibia portion. Zhu et al. [104] reported that GSH was more abundant in cartilage than in the meniscus or infrapatellar fat pad, although this cartilage was more susceptible to age-related GSH oxidation in aged OA rats. Carlo et al. [126] found that more chondrocytes from old donors died after exposure to SIN-1 (an oxidant) than those derived from young donors, and the activity of antioxidant enzymes was decreased in the older cells. These results provide evidence indicating that increased oxidative stress with ageing renders chondrocytes more susceptible to oxidant-mediated cell death through the dysregulation of the GSH antioxidant system. Furthermore, in addition to focusing on iron dyshomeostasis and lipid peroxidation, the kinetics of iron deposition, such as changes in different states (exercise, rest, etc.), including dynamic rhythm changes in iron-regulating proteins, need to be considered [127].

Currently, this encouraging evidence has generated high interest in further exploring the mechanisms underlying ferroptosis and OA. However, recent research on ferroptosis has been focused only on chondrocytes, ignoring cellular interactions and crosstalk, even though OA is a whole-joint disease involving cartilage, synovium, subchondral bone, and the infrapatellar fat pad. In contrast, evidence of lipid peroxidation has been found in articular synoviocytes. Rabbit synoviocytes induced with IL-1β or lipopolysaccharide treatment showed increased MDA levels, which were reversed by Ayurvedic drugs [128]. Yang et al. [129] found that GSH enhanced the antioxidant capacity of hyaluronic acid and modulated the expression of proinflammatory cytokines in human fibroblast-like synoviocytes induced by IL-1β. A combination of ascorbic acid and Fe^2+^ induced the production of radical-mediated lipid peroxidation in homogenates and/or the medium of cultured chondrocytes and synoviocytes, and the degree of lipid peroxidation in these chondrocytes was approximately threefold higher than that in synoviocytes [130]. These results suggest that ferroptosis may depend on the interaction between articular cells.

Furthermore, in addition to iron, other trace elements, such as copper or zinc, may be associated with OA progression [131]. A Mendelian randomisation study suggested that genetic predisposition to physiologically higher levels of circulating copper and zinc may increase the risk of OA [40]. The positive or negative correlations of trace elements in synovial fluids in patients with OA indicate a role played by these elements in OA development [132]. Interestingly, copper or zinc can participate in redox reactions and ferroptosis [133,134]. Therefore, attention to the interaction of various trace elements in ferroptosis may contribute to a better understanding of the role played by ferroptosis in OA.

Since ferroptosis is involved in the progression of OA, the regulation of iron homeostasis and the control of lipid peroxidation provide therapeutic options for OA. The iron chelator DFO [9,57,60,83,88,116] and the antioxidants Fer-1 [8,9,71,82,83,117] and CoQ10 [119,120] have shown significant anti-OA effects both in vitro and in vivo. This anti-OA effect has also been demonstrated in vitro by the calcium chelator BAPTA acetoxymethyl ester [86], the natural iron chelator lactoferrin [89], and the antioxidants NAC [101,116] and vitamin E [109]. An increasing number of agents, such as platelet-rich plasma [78], nifedipine [94], and icariin [97], have also been studied. However, given that intervention is always short-term and single-factor (single agent or single mode of administration) in in vitro studies, whereas the period is long in in vivo studies, factors such as pharmacokinetics and interactions with the molecules and cells of the body need to be considered [135]. Therefore, the efficacy and safety of both iron chelators and antioxidants deserve more research in vivo and in clinical situations.

## 7. Conclusions

In summary, as a newly described type of cell death, ferroptosis is closely associated with OA and may play an important role in OA occurrence and development. The regulatory mechanism of ferroptosis in OA and effective methods to regulate ferroptosis need to be urgently explored to provide a theoretical basis for the prevention and treatment of OA.

## Figures and Tables

**Figure 1 antioxidants-11-01668-f001:**
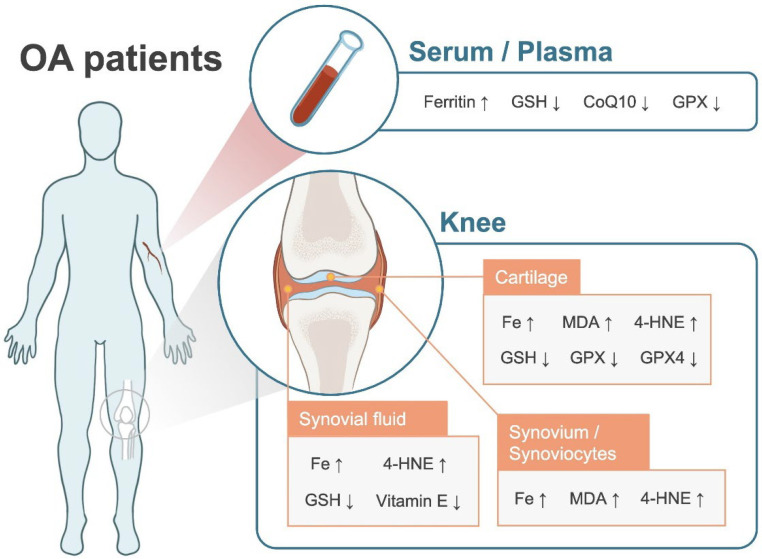
Iron dyshomeostasis and lipid peroxidation in OA patients. 4-HNE: 4-hydroxynonenal; CoQ10: Coenzyme Q10; GPX: Glutathione peroxidase; GSH: Glutathione; MDA: Malondialdehyde; OA: Osteoarthritis; ↑ indicates increased levels; ↓ indicates decreased levels.

**Figure 2 antioxidants-11-01668-f002:**
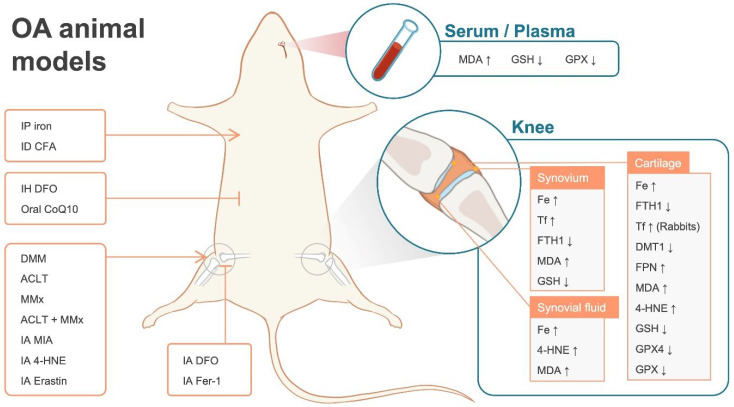
Iron dyshomeostasis and lipid peroxidation in OA animal models. 4-HNE: 4-hydroxynonenal; ACLT: Anterior cruciate ligament transection; CFA: complete Freund’s adjuvant; CoQ10: coenzyme Q10; DFO: deferoxamine; DMM: destabilization of the medial meniscus; DMT1: divalent metal-ion transporter 1; Fer-1: ferrostatin-1; FPN: ferroportin; FTH1: ferritin heavy chain 1; GPX: glutathione peroxidase; GSH: glutathione; IA: intra-articular injection; ID: intradermal injection; IH: hypodermic injection; IP: intraperitoneal injection; MDA: malondialdehyde; MIA: monosodium iodoacetate; MMx: medial meniscectomy; Tf: transferrin; ↑ indicates increased levels; ↓ indicates decreased levels; → indicates promotion of OA; ┤ indicates prevention of OA.

**Figure 3 antioxidants-11-01668-f003:**
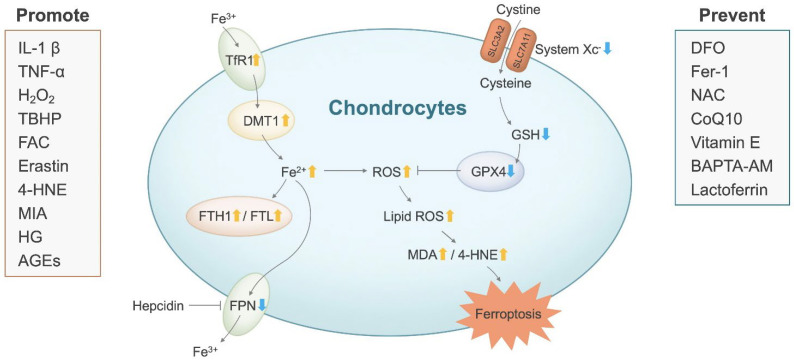
Iron dyshomeostasis and lipid peroxidation in OA chondrocytes. 4-HNE: 4-hydroxynonenal; AGEs: advanced glycation end products; BAPTA-AM: BAPTA acetoxymethyl ester; CoQ10: coenzyme Q10; DFO: deferoxamine; DMT1: divalent metal-ion transporter 1; FAC: ferric ammonium citrate; Fe^2+^: ferrous iron; Fe^3+^: ferric iron; Fer-1: ferrostatin-1; FPN: ferroportin; FTH1: ferritin heavy chain 1; FTL: ferritin light chain; GPX4: glutathione peroxidase 4; GSH: glutathione; H_2_O_2_: hydrogen peroxide; HG: high glucose; IL-1β: interleukin-1beta; lipid-ROS: lipid reactive oxygen species; MDA: malondialdehyde; MIA: monosodium iodoacetate; NAC: N-acetyl cysteine; OA: osteoarthritis; ROS: reactive oxygen species; SLC3A2: solute carrier family 3, member 2; SLC7A11: solute carrier family 7, member 11; TBHP: tertiary butyl hydroperoxide; TfR1: transferrin receptor 1; TNF-α: tumor necrosis factor- alpha; ↑ indicates increased levels; ↓ indicates decreased levels; Promote: agents to promote OA; Prevent: agents to prevent OA.

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
