# Peer review of "The Role Played by Ferroptosis in Osteoarthritis: Evidence Based on Iron Dyshomeostasis and Lipid Peroxidation"

_antioxidants, 2022, doi:10.3390/antiox11091668_

Round 1

Reviewer 1 Report

Given the novel subject matter covered in this review, I think the manuscript could benefit from a schematic diagram demonstrating how ferroptotic pathways are specifically involved in OA pathogenesis.  

Reviewer 2 Report

In the review article, the authors address the role of ferroptosis in osteoarthritis. The review article provides an overview of the current state in clinical, animal, and cell culture studies. Personally, I miss clear illustrations that illustrate the text and thus visually reinforce the knowledge. At least two of these should be included to enhance the review article.

Minor comments:

- Abbreviations should be avoided if possible to increase readability (e.g. knee OA, or abbreviations that are only used once).

- What is meant by “Patients” in Table 2? Please specify.

- Table 3 is too large and confusing and should be revised.

Reviewer 3 Report

This is a very good, comprehensive, explicit, and well written review describing the role of ferroptosis in the development and progression of OA. The authors included the essential details for the different molecular mechanisms that lead to iron dyshomeostasis and lipid peroxidation which result in ferroptosis of chondrocytes in OA. Importantly, they included evidence form human, animal as well as in vitro studies to provide an overall picture of this procedure.

Some minor points will improve the quality of the manuscript which is publishable.

1.       Lines 28-29. What does etc mean? Rephrase, add the appropriate components (e.g. meniscus) and also add references.

2.       Line 33. What about GAGs such as HA?

3.       It would be helpful to produce a detailed coloured illustration which will graphically include all the important information from Section 2 with regards to molecular pathways and iron homeostasis, the role of the main mediators and how this is dysregulated in OA.

4.       In Section 2, the authors need to briefly explain how iron reaches cartilage which is an avascular tissue.

5.       In Table 2, the specific mouse strain for each study has to be included since there are important differences between mouse strains in OA.

6.       Table 5 must be separated according to human or animal cell studies and presented as two distinct sets.

7.       A paragraph/section must be added to describe the different anti-ferroptotic approaches both in vivo and in vitro.

Reviewer 4 Report

The paper is a very well written and interesting review on the role of ferroptosis in osteoarthritis onset and progression.

The topic is updated and the review is well balanced and comphrensive 

Round 2

Reviewer 2 Report

The authors improved the manuscript a lot.

Reviewer 3 Report

The authors addressed the majority of my comments. A figure which will graphically illustrate the important molecular pathways and iron homeostasis, the role of the main mediators and how this is dysregulated in OA, as suggested in my previous review, would be helpful and it is not included in the revised form.